# Gut Microbiota and Predicted Metabolic Pathways in a Sample of Mexican Women Affected by Obesity and Obesity Plus Metabolic Syndrome

**DOI:** 10.3390/ijms20020438

**Published:** 2019-01-21

**Authors:** Alejandra Chávez-Carbajal, Khemlal Nirmalkar, Ana Pérez-Lizaur, Fernando Hernández-Quiroz, Silvia Ramírez-del-Alto, Jaime García-Mena, César Hernández-Guerrero

**Affiliations:** 1Departamento de Genética y Biología Molecular, Cinvestav-IPN, Av IPN 2508, Ciudad de México 07360, Mexico; alejandra.chavez@cinvestav.mx (A.C.-C.); nirmalkar@cinvestav.mx (K.N.); fernando.hernandez@cinvestav.mx (F.H.-Q.); 2Departamento de Salud, Universidad Iberoamericana, Ciudad de México, Paseo de la Reforma 880, Ciudad de México 01219, Mexico; anabertha.perez@ibero.mx (A.P.-L.); silvia.ramirezalto@gmail.com (S.R.-d.-A.)

**Keywords:** gut microbiota, Mexican women, metabolic syndrome, obesity, high-throughput DNA sequencing, ion torrent, 16S rDNA

## Abstract

Obesity is an excessive fat accumulation that could lead to complications like metabolic syndrome. There are reports on gut microbiota and metabolic syndrome in relation to dietary, host genetics, and other environmental factors; however, it is necessary to explore the role of the gut microbiota metabolic pathways in populations like Mexicans, where the prevalence of obesity and metabolic syndrome is high. This study identify alterations of the gut microbiota in a sample of healthy Mexican women (CO), women with obesity (OB), and women with obesity plus metabolic syndrome (OMS). We studied 67 women, characterizing their anthropometric and biochemical parameters along with their gut bacterial diversity by high-throughput DNA sequencing. Our results indicate that in OB or OMS women, Firmicutes was the most abundant bacterial phylum. We observed significant changes in abundances of bacteria belonging to the Ruminococcaceae, Lachnospiraceae, and Erysipelotrichaceae families and significant enrichment of gut bacteria from 16 different taxa that might explain the observed metabolic alterations between the groups. Finally, the predicted functional metagenome of the gut microbiota found in each category shows differences in metabolic pathways related to lipid metabolism. We demonstrate that Mexican women have a particular bacterial gut microbiota characteristic of each phenotype. There are bacteria that potentially explain the observed metabolic differences between the groups, and gut bacteria in OMS and OB conditions carry more genes of metabolic pathways implicated in lipid metabolism.

## 1. Introduction

Obesity (OB) is defined as abnormal or excessive fat accumulation that predisposes the affected individual to other diseases [1]. Some comorbidities commonly associated with each other are glucose intolerance, dyslipidemia, hypertension, type 2 diabetes, kidney failure, sleep apnea, and osteoarthritis [2]. The worldwide prevalence of obesity is constantly increasing. In women, for instance, the prevalence was 6.4% in 1975 and increased to 14.9% in 2014, more than doubling in four decades [3]. Mexico’s 2016 official government health and nutrition survey reported that, in general, seven of 10 adults were overweight or obese, and the prevalence was higher in women (75.6%) than in men (69.4%) [4].

The etiology of obesity is multifactorial. Variables such as an imbalance between energy intake and energy expenditure, psychosocial aspects, composition of gut microbiota, and genetic characteristics participate in the development of obesity and its principal comorbidities [5]. The energy imbalance in people has been detonated by means of a change in the lifestyle of global societies towards “the Western model,” which entails an increased consumption of fat, simple carbohydrates, and sugar, coupled with decreased physical activity. The altered nutrition of people with obesity as the consumption of fat and sugar increases is a pivotal factor in the alteration of the composition of gut microbiota, which causes dysbiosis [6].

The human gut microbiota is the complex and collective microbial community, mostly bacteria, that inhabits the human gut [7]. In the last decade, the gut microbiota has been recognized as an important contributor to obesity and other metabolic diseases [8], such as type 2 diabetes, non-alcoholic fatty liver disease, dyslipidemia, and metabolic syndrome [6].

In healthy people, the gut microbiota consists of five dominant bacterial phyla: Firmicutes, Bacteroidetes, Actinobacteria, Proteobacteria, and Verrucomicrobia, and less abundant phyla such as Cyanobacteria, Fusobacteria, and others [9,10]. Thus, the gut microbiota has been proposed as an environmental factor responsible for the control of body weight and energy metabolism, with a connection to obesity [10].

Obesity has been associated with changes in the phylum abundances of the gut microbiota [11], mainly Firmicutes and Bacteroidetes. However, the role of the ratio of Bacteroidetes to Firmicutes in obesity is still a controversial issue; it is much more complex than a simple difference in the relative abundance between phyla [12]. It has been suggested that the critical influence of some bacteria in the human metabolism may reside in the number of less abundant bacteria [13].

Metabolic syndrome (MS) is a disease with a prevalence of 20–25% in the adult population around the world [14]. In Mexico City, according to the Adult Treatment Panel III criteria, the reported prevalence for MS was 27% [15]. According to the American Heart Association (AHA) and the National Heart, Lung, and Blood Institute (NHLBI), MS is diagnosed when three or more risk factors concur. Such factors are abdominal obesity, high blood pressure, high blood levels of triglycerides, high fasting glucose, and low blood levels of High Density Lipoprotein-cholesterol (HDL-cholesterol), as well as a history of diabetes or antidiabetic, antihypertensive, or lipid-lowering medications [16].

Studies on MS and gut microbiota in mouse models have shown that the development of obesity and MS is the result of interactions between the gut microbiota, host genes, and diet [17]. In a study on gut microbiota and various glucose metabolism disorders in 20 Austrian men and women, it was found that some bacteria of the family Erysipelotrichaceae and Lachnospiraceae were associated with the metabolic disorder [18].

Gut microbiota dysbiosis can promote in diverse ways the occurrence of MS, such as low-grade inflammation by means of an increased production of bacterial lipopolysaccharide. Dysbiosis also promotes the synthesis of short-chain fatty acids (SCFAs), which affect the production of cholesterol and fatty acids in the liver, thereby altering the metabolism of lipids [19]. Likewise, SCFAs stimulate host energy harvesting and energy absorption, promoting an augmentation of body fat and body mass index (BMI). Gut microbes produce choline, betaine, and trimethylamine *N*-oxide metabolites from dietary phosphatidylcholines, which all participate in the development of atherosclerosis [20].

There are reports on gut microbiota and metabolic syndrome in relation to diet, probiotic or drug interventions, study of inflammatory pathways, environmental chemicals, ligand receptor interactions, economic status, and even host genetics in other populations [21]; however, it is necessary to explore the gut microbiota’s metabolic pathways in population like Mexicans, where the prevalence of obesity and metabolic syndrome is high. In this work, we studied the gut microbiota in a sample of volunteer Mexican women with CO, OB, or OMS. Our results show that the gut bacterial composition differed among these groups. Our findings may contribute to the development of viable treatments for the management of OB and OMS through the modulation of the gut microbiota by diet, prebiotics, probiotics, or fecal transplantation.

## 2. Results

### 2.1. Mexican Women Who Suffer from OB or OMS Exhibit Clinical Parameters in Accordance with Their Condition

The total cohort was 67 women. The clinical characteristics of the subjects permitted us to classify them into three categories: 25 CO, 17 women who suffered from OB, and 25 women who suffered from OMS. The anthropometric data show that, among the three groups, women with OB and OMS had the largest values in weight, body mass index, waist circumference, hip circumference, and waist-to-hip ratio (Table 1). In the same manner, the biochemical data show that women with OB and OMS had elevated fasting blood glucose, triglycerides, cholesterol, and low-density lipoprotein, and decreased high-density lipoprotein (Table 1). In addition, the blood pressure was normal in the CO and OB groups, and it was normal, on average, in the OMS condition (Table 1). A dietary assessment of macro- and micronutrient consumption showed no significant difference between the groups (Appendix A).

### 2.2. Firmicutes Is the Most Abundant Bacterial Phylum in Women Who Suffer from OB or OMS

We characterized the gut bacterial diversity in fecal samples of the study subjects as described in the Section 4.7. We found that in all three groups, the dominant phyla from the highest to the lowest percentage of relative abundance was Firmicutes, Bacteroidetes, Proteobacteria, and Actinobacteria (Figure 1). The statistical analysis by the Kruskal‒Wallis test (*p* = 0.0029) and Benjamini‒Hochberg correction (*q* = 0.0345) indicated a significant difference in the relative abundance of Firmicutes, with 56.95% for the CO group, 72.97% for the OB group, and 73.34% for the OMS group (Figure 1 and Appendix A). However, there was not a significant change in the relative abundance of Bacteroidetes (*p* = 0.7125, *q* = 1.0000), although the CO group had a relative abundance of 36.20%, the OB group 22.50%, and the OMS group 23.43%. The other phyla (Proteobacteria, Actinobacteria, Tenericutes, Cyanobacteria, and Synergistetes) were also not different between groups (Figure 1 and Appendix A). The “Others” category, which included less abundant phyla such as Verrucomicrobia, Spirochaetes, and Fusobacteria, showed a significant difference between groups (*p* < 0.0001, *q* < 0.0001) (Figure 1 and Appendix A). The bacterial load and relative abundance of Firmicutes and Bacteroidetes were also assessed by qPCR as described in the Section 4.10. The results showed that there is not a significant change in the bacterial load (*p* = 0.646), and the Bacteroidetes abundance (*p* = 0.661); however, the data suggested a slight increase in the Firmicutes abundance (*p* = 0.053) among the groups (Appendix A). A further analysis of the data, using the Tukey test as a post hoc test, confirmed that there is at least an increase in Firmicutes (*p* = 0.046) in the OMS group with respect to the CO group.

### 2.3. The Relative Abundance of Some Bacteria Changed in Women with OB or OMS

At the genus level, the statistical analysis indicated a significant decrease in the relative abundance of Bacteroides (*p* < 0.0001, *q* = 0.0002) a member of the phylum Bacteroidetes and the family Bacteroidaceae, in OB and OMS conditions in comparison with the CO groups. For the phylum Firmicutes, *Faecalibacterium* (*p* = 0.0003, *q* = 0.0290), a member of the Ruminococcaceae family, increased 2-fold from 0.55% in the CO group to 1.15% in the OB group and 1.19% in the OMS group. We also observed a significant increase of at least 2.8-fold in the relative abundance of members of the Lachnospiraceae family. In this family, *Roseburia* (*p* = 0.0002, *q* = 0.0231) had an abundance of 0.89% in the CO group, 2.72% in the OB group, and 2.14% in the OMS group, and *Lachnospira* (*p* < 0.0001, *q* = 0.0075) had an abundance of 0.99% in the CO group, 3.24% in the OB group, and 3.79% in the OMS group. Finally, *Coprococcus*, (*p* = 0.0002, *q* = 0.0231) had an abundance of 2.18% in the CO group, 4.55% in the OB group, and 4.51% in the OMS group. Remarkably, the family Erysipelotrichaceae (*p* < 0.0001, *q* = 0.00075) had an almost 5-fold decrease in abundance from 1.74% in the CO group to 0.38% in the OB group and 0.36% in the OMS group. All comparisons refer to the three groups CO, OB, and OMS, and a Kruskal‒Wallis test was used to calculate the *p*-value and the False Discovery Rate (FDR-adjusted) *q*-value, as shown in the Appendix A and Appendix A.

We did association analyses of the bacterial relative abundances with the clinical parameters using multivariate analyses with MaAsLin as indicated in Section 4.8. We only found positive association of the genus *Bilophila* with the body weight for all samples (Appendix A, *p* = 1.62 × 10^−6^, *q* = 0.00401), being the only variable with association between all clinical parameters and all macro/micronutrients. In addition, same genus was more abundant in OB in comparison to OMS (Appendix A, *p* = 0.000288, *q* = 0.237).

### 2.4. The Gut Microbiota Alpha Diversity of Women with OMS Is Higher than the Diversity in OB and CO

We estimated the alpha diversity in the CO, OB, and OMS conditions and made *t*-tests in pairs of groups to find statistically significant differences. We found differences in Chao1 index between CO and OMS groups (*p* = 0.003), and CO and OB groups (*p* = 0.002) (Figure 2 and Appendix A). There is a trend for Observed species and Chao1 index indicated that the bacterial community in the OMS group had the greatest richness, followed by the OB and CO groups. The Shannon index, an estimator of diversity, was higher in the OB condition than in the OMS or CO conditions. Moreover, the Simpson index, another indicator of diversity, showed that the OB bacterial community was more diverse than the communities found in the OMS and CO conditions.

### 2.5. The Beta Diversity of the Gut Microbiota of Women with OB or OMS Is Similar to Each Other but Different from the CO Group

The beta diversity was calculated using weighted and unweighted UniFrac analysis and distances estimated among samples from the three groups. For the unweighted, the three-dimensional scatter plot generated using principal coordinate analysis (PCoA) clearly grouped the OB and OMS gut microbiota communities, separating them from the CO gut microbiota (Analysis of Similarities, ANOSIM; *p* = 0.01) (Figure 3). A similar result was observed for the weighted analysis (ANOSIM, *p* = 0.01) (Appendix A and Appendix A).

### 2.6. Gut Bacteria from 16 Different Taxa Might Explain the Observed Metabolic Differences Among the CO, OB, and OMS Groups

Based on the relative abundances of the gut bacteria in the CO, OB and OMS groups, we used a linear discriminant analysis effect size (LEfSe) algorithm, followed by Benjamini‒Hochberg correction (*q*-values), to determine the bacteria that might explain the observed metabolic differences between the groups (Figure 4). Six genera and families (Lachnospiraceae, *p* = 0.0001, *q* = 0.0048; *Lachnospira*, *p* < 0.0001, *q* = 0.0011; *Coprococcus*, *p* = 0.0001, *q* = 0.0032; *Faecalibacterium*, *p* = 0.0002, *q* = 0.0063; *Ruminococcus*, *p* = 0.0001, *q* = 0.0048; and *Megamonas*, *p* < 0.0001, *q* = 0.0001) were enriched in the OMS group (green color), while seven genera and families (*Bacteroides*, *p* < 0.0001, *q* = 0.0005; *Streptococcus*, *p* = 0.0010, *q* = 0.0285; Erysipelotrichaceae, *p* < 0.0001, *q* = 0.0017; *Parabacteroides*, *p* = 0.0009, *q* = 0.0272; *Staphylococcus*, *p* < 0.0001, *q* = 0.0011; *Turicibacter*, *p* < 0.0001, *q* = 0.0006; and *Lactococcus*, *p* = 0.0006, *q* = 0.0199) were enriched in the CO group (red color), and three genera and families (*Roseburia*, *p* = 0.0002, *q* = 0.0063; *Succinivibrio*, *p* < 0.0001, *q* < 0.0001; and S24–7, *p* < 0.0001, *q* < 0.0001) were enriched in the OB group (blue color) (Appendix A).

### 2.7. The Predicted Functional Metagenome of the Gut Microbiota Shows Differences among the Groups

Next, we explored if the metabolic pathways found in the gut microbiota were related to the metabolic differences found in the CO, OB, and OMS groups. We performed a comparative prediction analysis of the functional metagenome (PICRUSt) of the gut bacterial microbiota as described in Section 4.9. We identified 16 metabolic pathways where the difference in % of relative frequency was statistically significant among the three groups CO, OB, and OMS (Figure 5), with *p*-values ≤ 0.0178, and *q*-values ≤ 0.0293 (Appendix A). In addition, post hoc testing was used to determine where the statistically significant difference was between pairs of groups, and the results showed that metabolic pathways involved in glycerolipid metabolism (*p*-value = 0.001), lipid metabolism (*p*-value < 0.001), and the synthesis and degradation of ketone bodies (*p*-value = 0.017) were significantly increased in the gut microbiota of OMS individuals with respect to CO; metabolic pathways involved in glycerophospholipid metabolism (*p*-values ≤ 0.005) and panthothenate and CoA biosynthesis (*p*-values ≤ 0.009) were significantly increased in the gut microbiota of OMS and OB individuals with respect to CO, while metabolic pathways involved in alanine aspartate and glutamate metabolism (*p*-value = 0.001), energy metabolism (*p*-value = 0.033), glycolysis gluconeogenesis (*p*-value = 0.013), lipid biosynthesis proteins (*p*-value = 0.005), pyruvate metabolism (*p*-value = 0.015), and type II diabetes mellitus (*p*-value = 0.001) were significantly increased in the gut microbiota of CO individuals with respect to OMS, and metabolic pathways related to the adipocytokine signaling pathway (*p*-values ≤ 0.033), amino sugar and nucleotide sugar metabolism (*p*-values ≤ 0.037), glycosaminoglycan degradation (*p*-values ≤ 0.004), lipopolysaccharide biosynthesis (*p*-values ≤ 0.035), and taurine and hypotaurine metabolism (*p*-values ≤ 0.014) were significantly increased in the gut microbiota of CO individuals with respect to OB and OMS (Figure 5 and Appendix A).

## 3. Discussion

In this work, we studied Mexican women affected by obesity and obesity plus metabolic syndrome. Sixty-seven volunteer women were studied: 25 CO, 17 OB, and 25 OMS (Table 1). It is remarkable that OB and OMS women had almost double the average age of the CO group. The reason was that it was not possible to find older unaffected women in the community where the sample was collected. For instance, according to the most recent epidemiological survey made by the Mexican government in 2016, the prevalence of abdominal obesity was 75.6% in adult women [4].

Women in the OB and OMS groups had dyslipidemia, and while fasting glucose and total cholesterol were normal in OB, the OMS women had a fasting glucose level suggesting prediabetes [22]. The LDL-c/HDL-c ratio was significantly higher and very close to the 3.0 cutoff value for cardiovascular risk in the OMS and OB groups with respect to the CO group. The average TC/HDL-c ratio in the OMS group was higher than the 4.5 cutoff ratio, indicating cardiovascular risk for these women [23]. Other parameters, such as systolic and diastolic blood pressure, were in the normal range for Mexican women in the OB and OMS groups [24].

The characterization of the gut microbiota showed that Firmicutes were more abundant in women who suffered from OB or OMS than in women in the CO group (Figure 1 and Appendix A). This observation was confirmed by qPCR, at least for the OMS group, in which we also observed no difference in the bacteria load among the CO, OB, and OMS groups (Figure 1 and Appendix A). Dysbiosis of the gut microbiota involving bacteria belonging to the phylum Firmicutes has been reported in 58–71-year-old Austrian women [18]. Firmicutes is believed to contribute to the occurrence of obesity, and in our study, the ratio for the relative abundance of Firmicutes/Bacteroidetes for the OMS group was 3.13, for the OB group 3.24, and for the CO group 1.57. This result agrees with previous reports showing that a higher relative abundance of Firmicutes is associated with obesity in Mexican children aged six to 14 years [19]. In addition, although it was not statistically significant, the abundances of Bacteroidetes and Actinobacteria in our study were higher in women in the CO group than in women in the OB and OMS groups (Figure 1 and Appendix A); however, the relative abundance of the genus *Bacteroides* spp. was higher in the CO group than in the OB and OMS groups (*p* < 0.0001, *q* = 0.0002) (Appendix A and Appendix A). Bacteria of these phyla could inhibit the growth of Firmicutes [25].

We observed that the abundance of some genera differed in the OB and OMS groups compared to the CO group. A non-parametric statistical analysis of the relative abundances of bacteria between the groups (Appendix A) showed differences in members of three taxa. *Faecalibacterium* spp. of the Ruminococcaceae family was significantly more abundant in the OB and OMS groups (Appendix A). The abundance of *F. prausnitzii* in feces (an important butyrate producer) is correlated to obesity in some populations but not in others, suggesting that ancestry, geographic location, and diet should be considered while interpreting results in all studies [26]. On the other hand, *Roseburia* spp., *Lachnospira* spp., and *Coprococcus* spp. from the Lachnospiraceae family were significantly more abundant in the OB and OMS groups (Appendix A). This family has been associated with diet-related obesity in British men [27]. Gut members of the Lachnospiraceae family as well as the Ruminococcaceae degrade complex polysaccharides into short-chain fatty acids such as butyrate, propionate, and acetate [28]. The family Erysipelotrichaceae was significantly decreased in the OB and OMS groups (Appendix A); there are reports implicating this family in metabolic disorders such as obesity [29], but Erysipelotrichaceae decreases in the gut of individuals with obesity after gastric bypass [30].

The analysis of the bacterial richness and alpha diversity among the groups showed that women in the OB and OMS groups had more rich and diverse communities than did women in the CO group (Figure 2). In the same manner, when beta diversity was measured, the bacterial communities from the OMS and OB individuals grouped together, separate from the CO individuals (Figure 3). Our results contrast with the report that gut bacterial diversity was low in male obese subjects affected by metabolic syndrome but increased after allogenic or autologous gut microbiota transfer (which increased insulin sensitivity) [31], but our results agree with reports showing the bacterial diversity is higher in obese individuals than individuals with normal BMI [32]. However, there are other reports showing no difference in the bacterial diversity between non-obese and metabolic syndrome subjects [18].

Our results on bacterial diversity suggest that it is important to analyze the type and function of bacteria that compose the microbiota. To do this, we analyzed the high-throughput DNA sequencing data using the LEfSe algorithm, obtaining relevant bacteria that were significantly enriched in the CO, OB, and OMS groups (Figure 4). The results demonstrate that some bacteria potentially explain the observed metabolic differences between the groups (Appendix A). In this manner, we identified bacteria characteristic to every group in the study. The predicted functional metagenome analysis using PICRUSt and STAMP showed metabolic pathways present in the microbiota genomes that are more frequent or less frequent in the OMS and OB conditions in comparison to the CO (Figure 5). For instance, some metabolic pathways involve enzymes for lipid metabolism, ketone body synthesis, and CoA biosynthesis. On the other hand, CO women carried bacteria with more abundant metabolic pathways related to amino acids, amino sugars, and energy metabolism, as well as glycolysis/gluconeogenesis and adipocytokines (Appendix A). One possibility is that in each group, the host, through the diet, favors the proliferation of microbiota with selected metabolic pathways that interact syntrophically with other bacteria that influence the host metabolism to define the anthropometric and biochemical profiles found in those women (Table 1).

The gut microbiota of the OMS subjects was significantly enriched in members of the phylum Firmicutes. These bacteria produce SCFAs such as butyrate (*Faecalibacterium* spp., *Coprococcus* spp., Family Lachnospiraceae, and *Ruminococcus* spp.) and acetate (*Coprococcus* spp.). There were bacteria reported in cases of obesity (*Faecalibacterium* spp., Family Lachnospiraceae, *Megamonas* spp.) and cases of metabolic syndrome (Family Lachnospiraceae), and there were cases in which bacteria abundance decreased after gastric bypass surgery in other populations (*Coprococcus* spp.) (Table 2). In the OB group, the gut microbiota was significantly enriched in the phylum Proteobacteria, with examples reported in cases of overweight (*Succinivibrio* spp.). In the same group, among the phylum Bacteroidetes, there are reports of SCFA producers and lipopolysaccharides (LPS) producers (Family S24–7), and among the phylum Firmicutes, there are reports of diseases such as obesity and type 2 diabetes (*Roseburia* spp.) (Table 2). Finally, in the CO group, women had a gut microbiota significantly enriched in the phylum Bacteroidetes; this agrees with reports that this genus negatively correlates with energy intake and adiposity (*Bacteroides* spp.) and is significantly decreased in obesity (Bacteroidaceae family). The CO group also had a higher relative abundance of *Parabacteroides* spp., members of which attenuate experimental murine colitis (*Parabacteroides distasonis*). The Firmicutes phylum was also enriched in CO, with members of the genera *Streptococcus* spp., *Staphylococcus* spp., *Turicibacter* spp., *Lactococcus* spp., and the family Erysipelotrichaceae. Some of these bacteria have been associated with reduction of body weight (*Streptococcus thermophilus*) and immunomodulatory properties (*Streptococcus* spp.). In humans, other enriched bacteria have been associated with metabolic disorders (Erysipelotrichaceae family), have had higher relative abundance in metabolic syndrome in other populations (*Staphylococcus* spp.), and have shown a negative association with clinical indicators of metabolic disorders (*Turicibacter* spp.) (Table 2). We did additional association analyses of the bacterial relative abundances with the clinical parameters using multivariate analyses with MaAsLin; as results we only found positive association of the genus *Bilophila* with body weight for all samples being this genus more abundant in OB in comparison to OMS (Appendix A). However, more studies are needed to determine if *Bilophila* has a role in this type of patient.

Another possible explanation for our results is that changes in the microbial diversity are due to age, since it was not possible to find healthy older unaffected women in the community and the Control group included younger women. To verify this issue, we first made a beta-diversity analysis using an aged-matched smaller cohort of women with well-established phenotypic characteristics, grouped in three sub-sets of three control (22–29 years old), three obese (22–28 years old) and three obesity + metabolic syndrome (22–29 years old) women. We observed that separated clustering of OB and OMS individuals with respect to CO persisted, as was observed in the whole analysis (Appendix A and Appendix A).

We made additional analyses to detect statistically significant changes in the gut microbial diversity due to age. In one of them we grouped the study subjects into 18–30 and 31–59 years groups and performed a LEfSe analysis. Based on the results, significant enrichment of the genus *Succinivibrio*, and the family S24–7 for obesity, as well as the family Lachnospiraceae (genera *Lachnospira*, and *Coprococcus*) for obesity + metabolic syndrome, may be attributed to age (31–59 years old), while the enrichment of genus *Roseburia* for obesity, and genera *Ruminococcus* and *Megamonas* for obesity + metabolic syndrome are due to the metabolic condition and not age. We should mention that this conclusion is biased by the fact that in the 31–59 years group 40% of women suffer from obesity and 60% suffer from obesity + metabolic syndrome (Appendix A). In the other analysis we removed age as a confounding variable, exploring the significant enrichment of members of the gut bacterial community by age in the obesity and obesity + metabolic syndrome categories by separate. For obesity we divided the women into 22–34, 35–42, and 43–51 years groups; for obesity + metabolic syndrome we divided the women into 22–34, 35–44, and 45–59 years groups. We did not observe a statistically significant enrichment of bacteria as a consequence of age. We also performed multivariate association with linear models (MaAsLin, v0.0.4), seeking an association between age and gut microbiota; however, we found no association. Taking all these results together, our interpretation is that enrichment of the genus *Roseburia* for obesity, and genera *Ruminococcus* and *Megamonas* for obesity + metabolic syndrome are more likely due to metabolic condition and not age; while for the case of the family S24–7, and genus *Succinivibrio* for obesity, and family Lachnospiraceae (genera *Lachnospira* and *Coprococcus*), and genus *Faecalibacterium* for obesity + metabolic syndrome, are due to metabolic condition and age.

In humans, the importance of the role the gut microbiota plays in energy intake and harvesting from diet has been previously reported. It is currently accepted that there is an obesity-associated gut microbiota harvesting higher energy from soluble dietary fiber through fermentation, producing more short-chain fatty acids than are made in lean individuals, and influencing in this manner the host energy metabolism [39]. In addition, the gut microbiota and their metabolites are remarkably involved in host appetite control by the modulation of host satiety pathways [40]. In some cases, our results contrast with what is reported in the available literature; however, we hypothesized that in OB and OMS women the gut microbiota has increased saccharolytic activity, producing more short-chain fatty acids, increasing the levels of Acetyl-CoA in the host, and causing dyslipidemia due to elevation of cholesterol and triglycerides, as has been reported for obesity in Mexicans [41] and Italians [42].

## 4. Materials and Methods 

### 4.1. Study Subjects

The 67 volunteer women of this cross-sectional analytic study were recruited from among people attending the Nutrition Clinic at the Universidad Iberoamericana in Mexico City. The inclusion criteria were aged 18 to 59 years and had taken no antibiotic treatment in the three months before the study. The exclusion criteria were chronic diseases, smoking, pregnancy, allergies, thyroid disease, eating disorders, consumption of any supplement, and atherosclerotic cardiovascular disease [43]. Women were chosen in this study to avoid gender bias [44], and because the overweight and obese condition is higher in women (75.6%) than in men (69.4%) in the Mexican population [4]. We used the guidelines from the American Heart Association to diagnose metabolic syndrome [16], which must adhere to at least three of the five following parameters; 1. Elevated waist circumference: waist circumference of ≥80, 2. Elevated Triglycerides ≥150 mg/dL, 3. Reduced HDL- Cholesterol <50 mg/dL, 4. Elevated Blood pressure; systolic ≥130 mm Hg and/or diastolic ≥85 mm Hg and 5. Elevated Fasting blood glucose ≥100 mg/dL. On the other hand, the women with just OB were identified using body mass index (BMI) as follows: normal rank was considered 18.5–24.99 kg/m^2^, the rank for obese class I was 30–34.99 kg/m^2^, obese class II 35–39.99 kg/m^2^, and obese class III greater than or equal to 40 kg/m^2^. The participants were divided into three groups: 25 controls with normal weight (CO), 17 with obesity (OB), and 25 with obesity plus metabolic syndrome (OMS). All participants signed informed consent in accordance with the Helsinki Declaration revised in 2000. The ethic and scientific committee of the Universidad Iberoamericana Mexico City approved the ID# 2018-01 research protocol in 28 October 2013.

### 4.2. Anthropometric Evaluation

The percentage of fat mass and body weight were measured under fasting conditions using an InBody Model 720 (±0.1 kg accuracy, 250 kg capacity, Biospace Co., Seoul, Korea). The height was measured using a stadiometer (Seca^®^ Model 240; ±2 mm accuracy, Seca GmbH & Co., Hamburg, Germany) in the standing position after removing shoes. The waist was measured at the midpoint between the lower rib and iliac crest.

### 4.3. Dietary Assessment

A certificated dietitian made dietary assessments using a frequency questionnaire validated for the Mexican population. Items were encoded, transformed to macro- and micronutrient units, and used to calculate the daily intake using the Nutritional Vector Calculation System adjusted to the Mexican diet [45] (Appendix A).

### 4.4. Biochemical Studies

Peripheral blood samples were taken by standard venipuncture in 7-mL heparin tubes (Becton Dickinson, Franklin Lakes, NJ, USA). Plasma was obtained, aliquoted in 1.5-mL cap tubes, and stored at −78 °C. Total cholesterol, high-density lipoprotein (HDL), low-density lipoprotein (LDL), triglycerides, and glucose were measured in mg/dL using the Cholestech LDX^®^ system. LDL-c/HDL-c and TC/HDL-c lipoprotein ratios were calculated and evaluated using the cutoff point of primary prevention [23].

### 4.5. DNA Extraction from Feces

Fecal samples were collected in a sterile stool container, aliquoted, and stored at −78 °C. DNA was extracted from 0.15 g of feces using the ZR Fecal DNA MiniPrep™ (Zymo Research, Irvine, CA, USA). The quantity of purified coproDNA was measured by its 260/280 nm absorbance ratio using a NanoDrop Lite Spectrophotometer (Thermo Scientific, Waltham, MA, USA), and the quality was evaluated by electrophoretic fractionation in 0.5% agarose gels.

### 4.6. Construction of the V3-16S rDNA Library and High-Throughput DNA Sequencing

For each participant, PCR was made using 10 ng of fecal DNA as a template, V3-341F forward primer containing a Golay barcode of 12-bp with an adapter, an antisense primer V3-518R and PCR master mix, and finally obtained an amplicon of 263-bp containing the V3 region of the 16S rDNA. The thermocycler program was: 5 min at 95 °C; 25 cycles of (15 s at 94 °C, 15 s at 62 °C and 15 s at 72 °C); subsequently, it was 10 min at 72 °C. PCR amplification was performed using PCR GeneAmp System 2700 Thermal Cycler (Applied Biosystems, Waltham, MA, USA). For the preparation of the library, the same amount in mass of each sample was quantified in equivalent amounts to approximately 10 μg each, and for each mixture of libraries of the V3 16S rDNA region they were fractionated by electrophoresis in 2% agarose gel. The amplicon was cut from the gel and purified using the SV PCR Cleaning System and the system (Promega, Madison, WI, USA). The DNA concentration of each library was measured by a NanoDrop Lite spectrophotometer (Thermo Scientific). The concentration and average size of each amplicon of the library were calculated with an Agilent 2100 bioanalyzer as previously described [19].

We performed the sequencing using the Ion 316 Chip Kit v2 (Carlsbad, CA, USA) and the Ion Torrent PGM System (Guilford, CT, USA.). Reads were filtered by the PGM software (Guilford, CT, USA.) to remove low-quality and polyclonal sequences. During this process, sequences matching the 3′ adapter were automatically trimmed and filtered. Trimmomatic software (v0.36, Aachen, Germany) was used to demultiplex the sequenced data based on barcodes. All reads were trimmed to length 200 nt. Filtered data were exported as FASTQ files. FASTQ files were converted to FASTA files, and all demultiplexed files were concatenated into a single file in order to determine with an open reference the OTUs and using a 97% similarity using QIIME pipeline (v1.9.0) and Geengenes database v13.8 [19]. 16S rDNA sequence files and corresponding mapping files for all samples used in this study were deposited in the NCBI BioSample repository (Accession Number: PRJNA417691) and can be found in the following link: https://www.ncbi.nlm.nih.gov/bioproject/PRJNA417691.

### 4.7. Microbial Diversity Analysis

Alpha diversity was generated with the script alpha_rarefaction.py in QIIME, using the following alpha params metrics: Observed Species, Chao1, Shannon, and Simpson to obtain the respective indexes. We used compare_alpha_diversity.py script to compares alpha diversities based on a two-sample *t*-test a non-parametric method using the default number of Monte Carlo permutations to find different significances among groups. The corresponding graphics were generated by R software using the phyloseq and ggplot2 libraries [9]. The analysis of sequenced data for Beta microbial diversity was made using the beta_diversity.py script of QIIME using weighted and unweighted UniFrac analysis and distances estimated among samples from the three phenotypes, the three-dimensional scatter plot was generated using principal coordinate analysis (PCoA). To determine the significant differences between all groups and after that in a sub-sample controlled by age, we use the compare_categories.py script of QIIME, using a distance matrix as the primary input and mapping file. As a statistical method we used ANOSIM, a nonparametric test in which its significance was determined through 99 permutations [46].

### 4.8. Analysis of Enrichment of Bacteria in Each Study Group

We used linear discriminant analysis effect size (LEfSe) to detect significant enrichment of gut bacteria. The LEfSe analysis was done with the relative abundance information file of all bacteria obtained at OTU-level genus assignment. The bacteria were selected as subclasses and the volunteer groups (CO, OB, and OMS) as classes for analysis. We used the script *run_lefse.py* to obtain *p*-values for the factorial Kruskal‒Wallis test among classes and obtained *p*-values for the pairwise Wilcoxon rankings with an LDA score >3 for any significant taxa between subclasses. Only *p*-values < 0.05 were significant. After this step, we used the script *plot_res.py* to generate the figures [47]. Finally, we ran the Benjamini‒Hochberg correction with the *p.adjust()* function in R software v3.4.2 to minimize the false discoveries [48,49]. To obtain associations of gut bacterial populations with metadata, we performed Multivariate Association with linear Models (MaAsLin), the data processed included age, anthropometric, biochemical, clinical and dietary intakes. Results were analyzed for false discovery rate by Benjamini‒Hochberg tests where only *p*-values < 0.05 and *q*-values < 0.25 were considered significant for an association between metadata and taxonomy [9].

### 4.9. Predictive Functional Genes of the Gut Microbiota Using PICRUSt

The prediction of the functional genes in the gut microbiota of the CO, OB and OMS subjects was done using the protocol from PICRUSt. We used a closed-reference OTU table in a biom-format from the script *pick_closed_reference_otus.py* generated in QIIME. The taxonomy assignment was made with the reference sequences from Greengenes database v13.8 with a 97% similarity. After that, the OTU table was normalized with the PICRUSt workflow using the Langille Lab Online Galaxy Instance (http://galaxy.morganlangille.com) to obtain the final metagenome functional prediction from the Kyoto Encyclopedia of Genes and Genomes (KEGG) database at hierarchy level 1 pathways [50]. Finally, we used the Statistical Analysis of Metagenomic Profiles (STAMP) software v2.1.3 to analyze the PICRUSt-predicted metagenomes to obtain significant differences in the functional genes between the groups using the Kruskal‒Wallis test, and we used Storey’s FDR approach for multiple test correction [51].

### 4.10. qPCR Assessment of the Bacterial Load and Composition

The bacterial load and abundances of Firmicutes, Bacteroidetes, and Actinobacteria were estimated by qPCR. We randomly selected seven CO, five OB, and seven OMS samples by proportionate stratified sampling method, corresponding to 25% of each group. Each PCR reaction had 12.5 μL of SYBR Green qPCR Master Mix (2×) (Thermo Fisher, Waltham, MA, USA), 0.8 μL of each primer (0.3 μM concentration), 2.5 DNA quenching (equilibrated to 5 ng), and 8.4 μL of water, obtaining a final volume of 25 μL. Each reaction was performed in triplicate. The amplification conditions were the following: a thermal cycling of 5 min at 95 °C denaturation, 30 cycles of 95 °C for 15 s, 61.5 °C for 15 s, and 72 °C for 20 s for alignment and for 5 min at 72 °C for elongation. We used StepOne Real-Time PCR System (Thermo Scientific), primers (Firmicutes, Bacteroidetes, Actinobacteria, and Universal), and conditions developed for each bacterial taxon [52]. For each pair of primers, we determined the specificity of products and the absence of primer dimers using a melting curve analysis. The Ct was measured by triplicate for each pair of primers. We used an ANOVA test to detect significant differences in the Ct among the groups (Appendix A), and a Tukey test to identify specific significant differences between pairs of groups.

### 4.11. Statistical Analyses

The clinical characteristics were statistically analyzed using SPSS v24 (IBM, Armonk, NY, USA) with the Kruskal‒Wallis test for non-parametric data and ANOVA for parametric data. The relative abundances of relevant bacterial phyla, orders, families, and genera were also assessed. To detect the differences between gut microbiota at all levels between the OMS, OB, and CO groups, we analyzed the relative abundances (%) of the bacterial communities with the Kruskal‒Wallis test to calculate *p*-values and the false discovery rate (FDR) through Benjamini‒Hochberg correction to calculate *q*-values using the *p.adjust()* function in R software v3.4.2 (R Foundation for Statistical Computing, Vienna, Austria).

## 5. Conclusions

Our hypothesis in this work was that healthy Mexican women with normal weight, OB, and OMS have bacterial gut microbiota characteristic of each phenotype. Our study validated this hypothesis; moreover, we found results indicating that the predicted metabolic pathways are also different. The study of the colon microbiota in different populations continues to provide new knowledge about the composition, diversity, and metabolic pathways of bacteria. This knowledge can be translated into treatments for OB and OMS consisting of modulations of the gut microbiota. This work is the first report describing the composition of the intestinal microbiota in adult Mexican women with obesity or obesity plus metabolic syndrome.

## Figures and Tables

**Figure 1 ijms-20-00438-f001:**
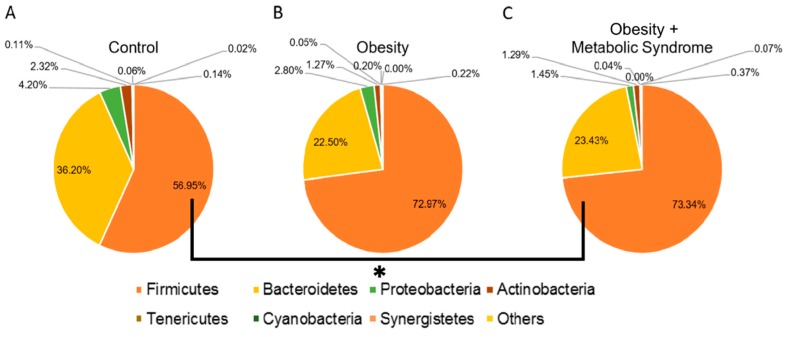
Bacterial phyla abundance. The figure shows circle charts of the relative abundance of relevant bacterial phyla in the three phenotypic categories. (**A**) Control; (**B**) obesity, and (**C**) obesity + metabolic syndrome. Phyla are identified by colors as indicated underneath the charts. Others includes phyla such as Verrucomicrobia, Spirochaetes, and Fusobacteria. Numbers are the relative abundance in percentage. Solid line and asterisk indicate a significant difference in the relative abundances for the phylum Firmicutes between the control and obesity + metabolic syndrome.

**Figure 2 ijms-20-00438-f002:**
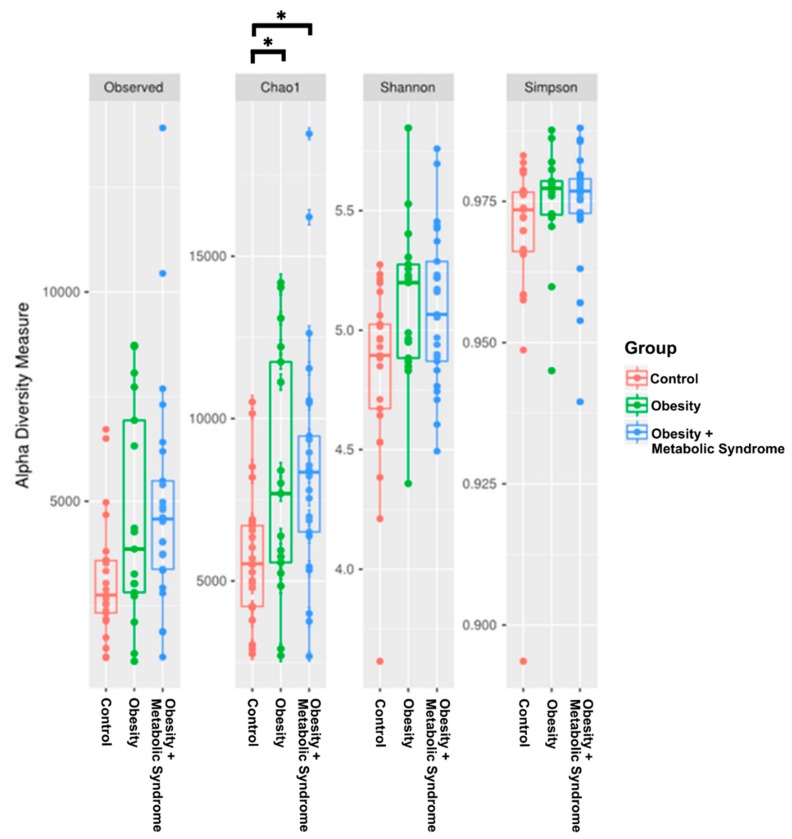
Bacterial alpha diversity. The box-plot figures show the alpha diversity of the bacterial communities in the three study groups Control, OB, and OMS by means of observed Operational Taxonomic Units (OTUs), and Chao1, Shannon, and Simpson indexes. Plotted in the graphic are the interquartile ranges (IQRs) and boxes, medians (lines in the boxes), and lowest and highest values for the first and third quartiles. Each phenotypic category is identified by colors, as indicated on the right side of the figure. Every sample is represented by a colored dot. Solid lines and asterisks indicate a significant difference between control and obesity (lower), (*p* < 0.002), and control and obesity + metabolic syndrome (upper) (*p* < 0.003). *P*-values were calculated to compares alpha diversities based on a two-sample *t*-test using a non-parametric methods and the default number of Monte Carlo permutations in order to find different significances among groups (see Appendix A).

**Figure 3 ijms-20-00438-f003:**
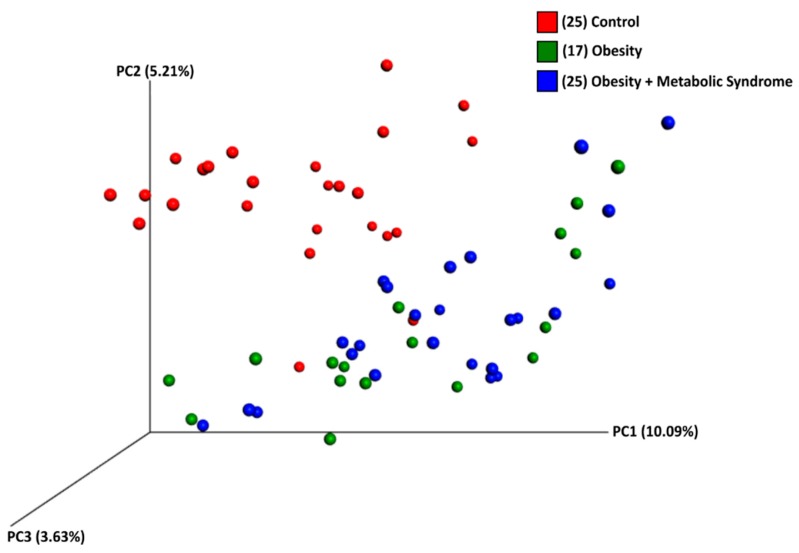
Bacterial beta diversity. The Figure shows a three-dimensional scatter plot, generated using principal coordinates analysis (PCoA) from Unweighted UniFrac analyses, showing the distance of microbial communities among women with normal weight (red spheres), women with obesity (green spheres), and obesity + metabolic syndrome (blue spheres). Each group is identified by colors as indicated on the right side of the figure. The *p*-value was calculated using ANOSIM method to compare beta diversities between each category in all samples, and between each category in a sub-sample controlled by age, using a distance matrix as the primary input and mapping file. Control, obesity, and obesity + metabolic syndrome have significant differences in all samples (*p*-value 0.01), and in a sub-sample controlled by age (*p*-value 0.02; see Appendix A).

**Figure 4 ijms-20-00438-f004:**
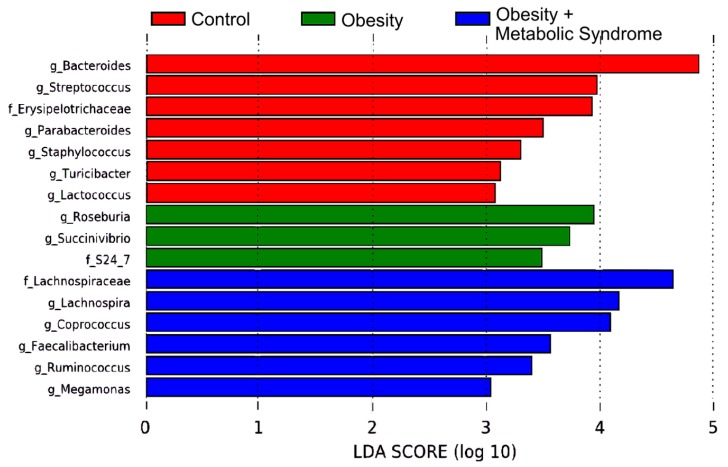
Linear discriminant analysis Effect Size (LEfSe) for the bacterial communities. The LEfSe plot shows enriched bacterial families and genera significantly associated with the three phenotypic categories. Seven bacteria were enriched in the control group (red), three bacteria in the obesity group (green), and six bacteria in the obesity + metabolic syndrome group (blue). The Linear Discriminant Analysis (LDA) score or effect size is shown at logarithmic scale underneath the bars. Each group is identified by color on top of the figure.

**Figure 5 ijms-20-00438-f005:**
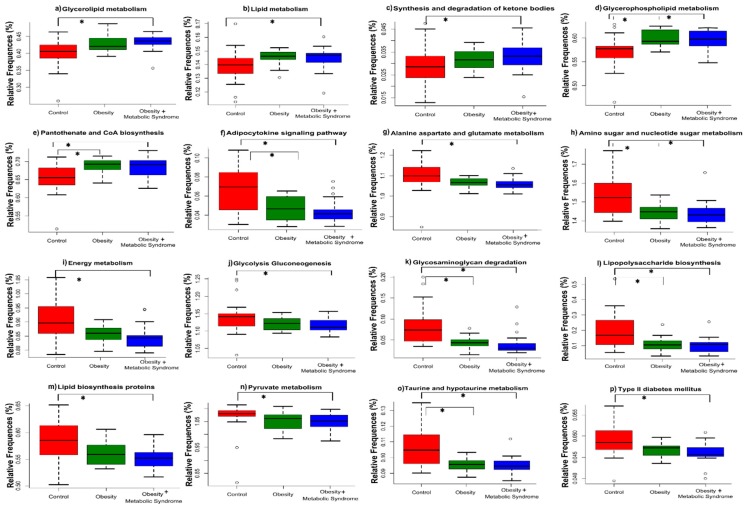
Comparative prediction of the functional metagenome of the gut bacterial microbiota. The figure shows a graphic representation of the significant predicted metabolic pathways using PICRUSt by analysis of the corresponding OTU table generated by QIIME for the bacterial communities. The Y-axis shows the relative frequencies of gene content prediction, and the X-axis shows the control (red), obesity (green) and obesity + metabolic syndrome (blue) categories. *, indicates significant difference among groups (see Appendix A).

**Table 1 ijms-20-00438-t001:** Clinical characteristics of the studied subjects.

	Control	Obesity	Obesity + Metabolic Syndrome	*p*-Value
Number of subjects	25	17	25	
Age (years)	23.3 ± 3.1	38.8 ± 8.4 *	40.5 ± 10.3 *	<0.001
Age range	18 to 30	22 to 51	22 to 59	nd
**Anthropometric data**				
Height (m)	1.61 ± 0.05	1.55 ± 0.08 *	1.56 ± 0.06 *	0.002
Weight (kg)	55.3 ± 6.4	83.2 ± 14.1 *	86.8 ± 15.1 *	<0.001
BMI (kg/m^2^)	21.4 ± 1.9	34.8 ± 6.1 *	35.8 ± 5.1 *	<0.001
WC (cm)	71.4 ± 7.0	96.9 ± 11.1 *	99.6 ± 10.2 *	<0.001 ^a^
HC (cm)	92.0 ± 6.7	114.6 ± 11.1 *	115.7 ± 11.8 *	<0.001 ^a^
W/H ratio	0.78 ± 0.6	0.85 ± 0.08 *	0.86 ± 0.05 *	<0.001 ^a^
**Biochemical data**				
Fasting Glucose (mg/dL)	84.2 ± 5.5	91.6 ± 05.6	104.6 ± 17.5 *	<0.001
Triglycerides (mg/dL)	109.4 ± 29.1	102.2 ± 27.0	177.2 ± 69.6 *	<0.001
HDL (mg/dL)	60.6 ± 12.4	45.6 ± 11.5 *	42.4 ± 8.4 *	<0.001
LDL (mg/dL)	87.4 ± 24.2	102.0 ± 21.1	116.4 ± 23.3 *	<0.001 ^a^
Cholesterol (mg/dL)	170.0 ± 26.2	168.0 ± 26.2	194.3 ± 29.0 *	0.002
LDL-c/HDL-c ratio	1.5 ± 0.6	2.5 ± 1.0 *	2.9 ± 0.8 *	<0.001 ^a^
C/HDL-c ratio	2.9 ± 0.7	3.9 ± 1.1 *	4.7 ± 1.1 *	<0.001
**Blood pressure**				
SBP (mm Hg)	109.3 ± 10.7	122.7 ± 16.6 *	125.4 ± 13.1 *	<0.001
DBP (mm Hg)	71.9 ± 10.9	80.2 ± 13.7	85.5 ± 12.7 *	0.003

The results appear like average ± standard deviation. *p*-value was calculated according to the Kruskal‒Wallis test for non-parametric data except for ^a^ ANOVA for parametric data. WC—waist circumference, HC—hip circumference, W/H—waist to hip ratio, HDL—high-density lipoprotein, LDL—low-density lipoprotein, C/HDL—cholesterol to HDL-c ratio, SBP—systolic blood pressure, DBP—diastolic blood pressure, nd—not determined. *p* < 0.05 are considered statistically significant. *, indicates statistically significant differences with respect to the control group. The significant differences between pair of groups are shown in Appendix A.

**Table 2 ijms-20-00438-t002:** Gut bacteria with significant abundance change among study subjects.

Taxa	This Work	Other Reports	Reference
**Phylum Proteobacteria**			
*Succinivibrio* spp.	3-fold more abundant in OB than CO and OMS	*Succinivibrio* spp. and *Halomonas* spp. overrepresented in Brazilian subjects with overweight and omnivores, this group showed higher values of insulin and HOMA-IR and a worse lipid profile.	[33]
**Phylum Bacteroidetes**			
Family S24–7	3-fold more abundant in OB than CO and OMS	LEfSe analysis indicated that members of the LPS-producing family S24–7 were 5-fold more abundant in mice on a high-fat diet versus a high-fat diet plus capsaicin (potent anti-obesity function).	[34]
*Bacteroides* spp.	4-fold more abundant in CO than OB and OMS	In women from Austria, the relative abundance of *Bacteroidaceae* family (genera from *Bacteroides*) was significantly decreased in obese but not non-obese women.	[18]
*Parabacteroides* spp.	3-fold more abundant in CO than OB and OMS	Oral administration of *Parabacteroides distasonis* antigens attenuates experimental murine colitis through modulation of immunity and microbiota composition.	[35]
**Phylum Firmicutes**			
*Streptococcus* spp.	3-fold more abundant in CO than OB and OMS	*S. thermophilus*, a probiotic, is associated with reduction of body weight, fat accumulation, and fatty acid synthase activity in adipocytes in mice.	[36]
*Roseburia* spp.	3-fold more abundant in OB than CO and OMS	Commensal bacteria producing short-chain fatty acids, especially butyrate. It is associated with several diseases including irritable bowel syndrome, obesity, type 2 diabetes, nervous system conditions, and allergies.	[37]
Family Erysipelotrichaceae	3-fold more abundant in CO than OB and OMS	High relative abundances of Erispielotrichaceae have been reported in women from Austria with impaired fasting glucose vs. normal glucose.	[18]
		Strongest evidence for a role for Erysipelotrichaceae in human disease comes from studies related to metabolic disorders.	[29]
*Coprococcus* spp.	4-fold more abundant in OMS than OB and CO	*C. comes* abundance decreases along with BMI, serum triglycerides, cholesterol, and LDL-cholesterol in German adults affected with T2D/OB after Roux-en-Y gastric-bypass surgery.	[38]
Family *Lachnospiraceae & Lachnospira* spp.	4-fold more abundant in OMS than OB and CO	Lachnospiraceae were more abundant in Austrian subjects with impaired fasting glucose and impaired glucose tolerance, and in women with MS.	[18]
*Turicibacter* spp.	3-fold more abundant in CO than OB and OMS	*Turicibacter* spp. had a negative association with clinical indicators of metabolic disorder such as insulin and HbA1c in Austrian population.	[18]

CO, control group; OB, obesity group; OMS, obesity + metabolic syndrome group; T2D, type 2 diabetes; HbA1c, glycosylated hemoglobin; MUFA diet, canola oil-rich diet; HOMA-IR, Homeostatic model assessment insulin resistance. See Appendix A, and LEfSe results in Figure 4.

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
