# Peer review of "Gut Microbiota and Predicted Metabolic Pathways in a Sample of Mexican Women Affected by Obesity and Obesity Plus Metabolic Syndrome"

_ijms, 2019, doi:10.3390/ijms20020438_

Round 1
Reviewer 1 Report
Here, the authors characterized the gut microbiota composition and functional potential (through inferred metagenomics) in Mexican women affected by obesity and obesity+metabolic syndrome.
Their work remains descriptive but adds to the current literature on the role of the microbiota in metabolic disorders. It is sometimes in contradiction with some established characteristics, suggesting that there is still much work to be done.
The paper is overall well written but minor English edits are still needed.
The authors should further discuss/clarify the following:
I. Clinical parameters. Please, specify which clinical characteristics were used to stratify subjects and whether (and for which parameters) there was a difference between OB and OMS.
II. Taxonomic results. As the authors state, info at phylum level is not particularly relevant. Please, tone down/delete any overstatement related to subtle differences in phylum relative abundance (e.g. L248-249). Similarly, please tone down statements related to taxa that might explain/are related to the observed metabolic differences between CO, OB and OMS; indeed, the authors did not explore correlations between microbiota and clinical data, and it is unclear what are the “observed metabolic differences” (please, see point I).
III. Statistics. The authors forgot to report p values and tests used in paragraphs 2.4-2.7. Furthermore, in paragraph 2.3, please specify the tests used and what comparison the p values refer to.
IV. Discussion. The paper does not include a real discussion on the role of the gut microbiota and differentially abundant taxa in obesity and/or metabolic syndrome. What is known about the involvement of the microbiota in energy intake, appetite control and so on? Please, better argue. Furthermore, their data are sometimes in contrast to the available literature but no hypothesis is advanced; please, better discuss. In addition, the entire section should be revised for refs (e.g. no ref is cited from L290 to L311).
V. Methods. Please provide more details, including: i) PCR mixture, thermocycling and library preparation; ii) bioinformatics (data processing, filtering, beta diversity analysis, multivariate analysis with MaAsLin for age, etc.); iii) qPCR mixture and thermocycling.
Additional comments:
- Introduction. Please, better clarify the study rationale and the gaps of knowledge that this work aims to fill at the end of this section.
- L58: ref 9 on enterotypes is inadequate here.
- L135: the “relative” abundance. Please, add “relative” throughout the manuscript when appropriate.
- L137: please, specify if the relative abundance of Bacteroides is increased or decreased.
- L149: higher instead of richer.
- L150: alpha diversity instead of species richness (since different metrics were used).
- L159: what about the weighted UniFrac? I suggest the authors to show it as a supplementary figure.
- L167: discriminant instead of discriminative.
- Fig. 2: please add observed OTUs among metrics, and specify p values and test in the legend.
- Fig. 3: please specify the comparison the p value refers to (wasn’t OB and OMS vs CO?).
- Fig. 5: please add the label on y axis and unit of measure, and revise the legend by deleting the list of metabolic pathways.
- L252: taxa instead of families.
- L281-282: the authors did not demonstrate this. Please, rephrase.
- L384: 16S rRDNA.
Author Response
2019-01-16 the point-by-point response to Reviewer 1 are in the uploaded PDF file "Chavez-Carbajal et al IJMS_421932-Reviewer 1".

Reviewer 2 Report
In this manuscript, authors aimed to study alterations in gut microbiota of Mexican women having obesity (OB) or obesity with Metabolic Syndrome (OMS) compared with healthy controls (CO). To do this, they extracted fecal DNA and performed 16S rDNA sequencing followed by analysis of sequencing data. They found that OB and OMS groups had more bacterial diversity and richness compared with Controls. OB and OMS groups had more butyrate producing bacteria than Controls. Authors claim that the genes that are implicated in metabolic pathways that lead to the phenotype of OB and OMS come from the bacteria that are over represented in their gut bacteria (i.e butyrate producing bacteria). Authors have used strong statistical tools (QIIME, Lefse, STAMP and PICRUSt programs) to validate their hypothesis that different bacterial communities represent different phenotypes (CO,OB,OMS). Though, there are no age-comparable controls, but authors have used statistical analyses to prove that the dysbiosis was not age-related. However, this manuscript is not very well written and needs to be improved. Authors should also take care of following minor concerns.
Abstract should begin by briefly introducing Obesity and Metabolic syndrome and why looking at bacteria was important.
What was the rationale of using women only in this study?
A brief description of eligibility criteria of OB and OMS should be included in methods (4.1).
Author Response
2019-01-16 the point-by-point response to Reviewer 2 are in the uploaded PDF file "Chavez-Carbajal et al IJMS_421932-Reviewer 2".
